# Detecting Sepsis in Patients with Severe Subarachnoid Hemorrhage during Critical Care

**DOI:** 10.3390/jcm11144229

**Published:** 2022-07-21

**Authors:** Armin Niklas Flinspach, Jürgen Konczalla, Volker Seifert, Kai Zacharowski, Eva Herrmann, Ümniye Balaban, Elisabeth Hannah Adam

**Affiliations:** 1Department of Anaesthesiology, Intensive Care Medicine and Pain Therapy, University Hospital Frankfurt, Goethe University, Theodor-Stern Kai 7, 60590 Frankfurt, Germany; kai.zacharowski@kgu.de (K.Z.); elisabeth.adam@kgu.de (E.H.A.); 2Department of Neurosurgery, University Hospital Frankfurt, Goethe-University Frankfurt, 60528 Frankfurt, Germany; juergen.konczalla@kgu.de (J.K.); volker.seifert@kgu.de (V.S.); 3Department of Biostatistic and Mathematic Modeling, Goethe University, 60590 Frankfurt, Germany; herrmann@med.uni-frankfurt.de (E.H.); balaban@med.uni-frankfurt.de (Ü.B.)

**Keywords:** neurosurgery, sequential organ failure assessment scores, sepsis, subarachnoid hemorrhage, systemic inflammatory response syndrome, infection, pneumonia, cerebral vasospasm

## Abstract

**Introduction:** Sepsis and septic shock continue to have a very high mortality rate. Therefore, the last consensus-based sepsis guideline introduced the sepsis related organ failure assessment (SOFA) score to ensure a rapid diagnosis and treatment of sepsis. In neurosurgical patients, especially those patients with subarachnoid hemorrhage (SAH), there are considerable difficulties in interpreting the SOFA score. Therefore, our study was designed to evaluate the applicability of the SOFA for critical care patients with subarachnoid hemorrhage. **Methods:** Our retrospective monocentric study was registered (NCT05246969) and approved by the local ethics committee (# 211/18). Patients admitted to the Department of Neurosurgery at the Frankfurt University Hospital were enrolled during the study period. **Results:** We included 57 patients with 85 sepsis episodes of which 141 patients had SOFA score-positive results and 243 SIRS positive detections. We failed to detect a correlation between the clinical diagnosis of sepsis and positive SOFA or SIRS scores. Moreover, a significant proportion of sepsis that was incorrectly detected via the SOFA score could be attributed to cerebral vasospasms (*p* < 0.01) or a decrease in Glasgow Coma Scale (*p* < 0.01). Similarly, a positive SIRS score was often not attributed to a septic episode (49.0%). **Discussion:** Regardless of the fact that SAH is a rare disease, the relevance of sepsis detection should be given special attention in light of the long duration of therapy and sepsis prevalence. Among the six modules represented by the SOFA score, two highly modules were practically eliminated. However, to enable early diagnosis of sepsis, the investigator’s clinical views and synopsis of various scores and laboratory parameters should be highlighted. **Conclusions:** In special patient populations, such as in critically ill SAH patients, the SOFA score can be limited regarding its applicability. In particular, it is very important to differentiate between CVS and sepsis.

## 1. Introduction

Systemic infections, especially sepsis and septic shock, continue to have a very high mortality rate of up to 46% [1]. The primary goal of the last consensus-based sepsis guideline is to ensure a rapid diagnosis and treatment of sepsis. Corresponding to this goal, a new screening score (known as the qSOFA) has been included as part of the Sepsis 3 Guidelines published in 2016. Furthermore, a novel diagnosis defining score was implemented and defined as the sequential or sepsis related organ failure assessment score (SOFA score) [2]. This SOFA score predicting mortality comprises six organ-specific modules with a score from zero to four points (range: zero to twenty-four points) for the detection of organ dysfunction. As a result, the SOFA score has been used in place of the systemic inflammatory response syndrome (SIRS) score, which was previously used to define sepsis (Appendix A). These detectable organ dysfunctions should be significant for the diagnosis of sepsis if the score increases by two points or more.

The following six organ systems are observed in a corresponding module.

Circulatory function represented by systolic blood pressure and, if necessary, the amount of catecholamines required to maintain an adequate mean pressure;Cognitive function, as determined by the Glasgow Coma Scale (GCS);Liver function, as measured by using serum bilirubin levels;Kidney function, as measured by monitoring serum creatinine levels and urine flow;Lung function, as measured via the calculation of the respiratory quotient (arterial blood oxygen partial pressure/inspiratory oxygen fraction);Hematopoiesis displayed and based on the platelet count.

During the development of the guidelines, it was shown that the SOFA score is particularly suitable for the detection of systemic infection in patients in intensive care (ICU) units [2,3]. The new consensus definition claims to allow for an interdisciplinary applicability. However, neurosurgical patients, especially those patients with subarachnoid hemorrhage (SAH), have a considerable number of difficulties in interpreting the SOFA score [3]. SAH is divided into five severity grades according to the neurological compromise of the GCS, as well as following the recommendations of the World Federation of Neurosurgical Societies (WFNS). Beginning at level two, a direct effect on the GCS occurs as a simultaneous SOFA score marker. Furthermore, patients with severe SAH are frequently sedated and intubated, which further compromises the GCS. In this respect, an inadequate assessment of the neurological status for sepsis detection via the SOFA score marker has to be considered [4].

SAH has also been shown to exhibit a risk of cerebral vasospasm (CVS) in 21–37% of cases, depending on the severity of the condition. To reduce the risk of CVS-related severe secondary cerebral ischemia, a prolonged need for deep sedation (Richmond agitation and sedation scale [RASS] ≤−4) with associated lengthened time duration of ventilation is often necessary [5].

With the objective of risk reduction regarding CVS, patients can receive the calcium antagonist nimodipine. However, due to its regularly occurring hypotensive side effect, nimodipine may compromise the blood pressure values that are determined for the interpretation in the SOFA score [6].

In addition, the current standard of care includes an increase in the mean arterial pressure (MAP) to a level of approximately 110 mmHg by using vasopressors (hyperdynamic therapy) in cases of established CVS [5,7,8]. The use of high amounts of catecholamines implies that the SOFA module for the assessment of the patient’s circulatory function is also not appropriate in these cases. To date, the extent to which the SOFA Score can successfully be applied to patients with subarachnoid hemorrhage has not been investigated. However, the high degree of importance of the detection of sepsis in this cohort of patients should be emphasized, as this group faces a high risk of sepsis and septic shock with a high fatality rate [9,10,11]. Furthermore, preclinical aspiration is often observed during initial hemorrhage in SAH, and there is also a high risk of infection during the course of critical care treatment. Previous studies have emphasized the sepsis risk of SAH patients by detecting up to 50% of SIRS positive criteria and detectable bacteremia in up to 24% of patients [9,11,12,13]. In addition to the aforementioned impact concerning the SOFA score, central dysregulation may cause the frequently observed paroxysmal sympathetic hyperactivity (PSH). Among other symptoms of PSH, typical clinical symptoms include the presence of tachypnea, increased systolic blood pressure and central fever [14,15].

Thus, this study was designed to evaluate the applicability of the SOFA score as recommended by the Sepsis 3 guidelines for critical care patients with moderate to severe subarachnoid hemorrhage.

## 2. Materials and Methods

### 2.1. Study Design

The study was approved by the local Ethics Committee of the Goethe University Frankfurt, Germany (# 211/18). In this retrospective monocentric study, we analyzed the diagnostic performance of a positive SOFA score in adult SAH patients who were admitted to the Department of Neurosurgery. Patients were included between 1 June 2017 and 1 June 2018. The following inclusion criteria were used: critically ill patients suffering from moderate to severe subarachnoid hemorrhage (WFNS > 2). The study was registered to the Clinical Trials.gov Protocol Registration and Results System (NCT05246969).

### 2.2. Patient Population

We included all patients who were admitted to the Department of Neurosurgery at the Frankfurt University Hospital during the study period. All of the patients received intensive care in accordance with the relevant recommendations. No specific treatment protocols were defined, as these treatments were solely conducted at the discretion of the attending neurosurgeon or intensivist. The observation period began with the admission to the hospital and ended with death or discharge from the ICU.

### 2.3. Data Collection

Clinical data were continuously recorded by using the local patient data management system (PDMS; MetaVision 5.4, iMDsoft, Tel Aviv, Israel). We recorded demographic data, examination results, laboratory results and vital signs. From these data we calculated SOFA scores, quick sepsis-related organ failure assessment (qSOFA) scores and SIRS scores at admission for each patient. The SOFA and SIRS score were re-evaluated twice a day during the ICU stay.

Sepsis was defined by the SOFA developing working group and was consistent with recent major sepsis studies as a combination of intravenous antibiotic therapy and microbial sampling [3,16]. Sepsis detection was performed according to the definition that was applied by the research group for clinical detection. Correspondingly, physicians’ and nurses’ daily records were reviewed for suspected and diagnosed sepsis, as well as any newly obtained microbiological samples and new antibiotic prescriptions following the guideline definition.

### 2.4. Statistical Analysis

Continuous variables are presented as mean +/− standard deviation and categorical variables are presented as frequencies and percentages. Correlation analysis used nonparametric spearman correlation. The influence of factors or covariates and especially SOFA and SIRS scores on occurrence of the first and the second septic episode are analysed by a time-dependent cox regression. To assess sensitivity and specificity of predicting septic episodes over the time-course of the ICU stay, we used a Haegerty survival ROC approach with a span of 15 days for each evaluated time point. Furthermore, we used a logistic mixed model under consideration of patient random effects to assess factors influencing false positive predictions by the SOFA score. A two-sided *p* value of ≤0.05 was considered to be statistically significant. Statistical analyses were performed with SAS statistical software (version 9.4, SAS Institute, Cary, NC, USA), SPSS (IBM Corp., Version 26, Chicago, IL, USA) and R (R Foundation for Statistical Computing, Vienna, Austria. The R packages “risksetROC”, “survival” and “nlme” were used.

## 3. Results

During the observation period, we were able to include 57 of 1254 screened patients and obtain complete data for the study analysis (Figure 1). The patient demographics and distribution of sepsis episodes are presented in Table 1.

Among the treated patients, 51.7% (*n* = 30) were female. The mean observation time in the ICU was: 17 (±17.3) days, of which 85 sepsis episodes were detected in 46 (79.3%) patients.

The observed SAH severity grades according to World Federation of Neurosurgery, as well as the therapy causes compromise of the SOFA score, are shown in Figure 2.

According to the Sepsis 3 guidelines, we observed 141 SOFA score positive episodes and 243 SIRS positive days. Statistics revealed a significant correlation (*p* < 0.01) between the duration of ICU stay and the number of septic episodes. The clinically imposed mean duration of a septic episode was 4 (±1.8) days. We failed to detect any correlation between the clinical diagnosis of sepsis and the SOFA or SIRS scores that were used to define sepsis-3 (or sepsis-2). Figure 3 and Table 2 shows the specificity and sensitivity values of the SOFA and SIRS scores over the treatment period for the detection of initial sepsis and the rediagnosis of sepsis.

CVS was diagnosed in 40 patients who developed 48 septic episodes while suffering from vasospasms, and the mean duration of CVS was 11 (±10.0) days. A significant proportion of potential sepsis episodes that was incorrectly detected via the SOFA score (29 of 89 cases) could be attributed to CVS (*p* < 0.01). The use of catecholamines to increase MAP showed a significant correlation between the SOFA score and the false sepsis prediction (*p* < 0.01). The detection of successful sepsis episodes under CVS was achieved in 14 septic episodes by applying the SOFA score.

Among 49 endotracheal intubations, 8 intubations were performed during the first (prehospital) rescue phase, and 35 intubations were performed during the first day. A significant association between an intubation- and/or sedation-related (RASS ≤ −4) decrease in GCS and false sepsis episode prediction of SOFA (*p* < 0.01) was observed. Thirty-eight of the eighty-nine false-positive SOFA alarms were attributed to a drop in GCS (Table 3).

For the other organ modules of the SOFA, only isolated increases in scoring were noted. For example, when regarding renal function, a single SOFA point or more was awarded in only 5.6% (63 measurement points) of the assessments, which was attributable to three patients with pre-existing chronic renal insufficiency and a single case of acute renal failure. Additionally, liver function in terms of increased bilirubin levels resulted in the allocations of one point at 33 (2.99%) follow-up assessments. For hematologic function based on platelet count, a single or more SOFA points were scored at 55 (4.9%) follow-up points, most of which were associated with postintervention-associated blood loss.

The SIRS score had a value of more than or equal to two points, which was commensurate with a definitional positive detection of sepsis in 243 measurements. Among these, 119 (49.0%) measurements could not be attributed to a septic episode. The sensitivity and specificity of the SIRS score are shown in Figure 2 with respect to the ability to detect an initial septic episode, as well as a second sepsis episode. The majority of successful sepsis episode detections were based on the coincidence of leucocytosis and fever (24.3%), tachycardia (21.8%) or tachypnoea (36.4%).

## 4. Discussion

We observed a high rate of false diagnosis of sepsis when using the recommended SOFA score, which suggests that it may be particularly difficult to apply this score in critically ill SAH patients suffering from CVS. Regardless of the fact that SAH is a rare disease, it often affects patients abruptly and at a young age (as observed in our population). The relevance of sepsis detection in severe SAH should be given special attention, in light of the long duration of therapy and sepsis prevalence. To optimize treatment, the early detection of sepsis appears to be highly relevant, as in all critically ill patients. However, our data showed that, in addition to an expected significant correlation of the SOFA score to intubations with consecutive sedation and impairment of the GCS, other factors may also play a significant role. In particular, when considering mono-H therapy for CVS, we demonstrated that a significant proportion of false detections can be attributed to the assessment of the circulatory module by the necessity of catecholamine treatment. Coupled with the sedation-induced decrease in the Glasgow Coma Scale and the significant misdetection of sepsis, the ability to diagnose sepsis becomes increasingly difficult. Among the six modules represented by the SOFA score, two were practically eliminated. However, both modules are highly relevant for early sepsis detection, which is reflected by the modified use in the quick SOFA score, among others [3]. Although the trained intensivist was aware of the limitations of the GCS in ventilated patients, this factor is less considered in the other components of the SOFA score [17]. Moreover, when a sole focus on the SOFA score after several modules has been exhausted, thus no longer demonstrating an increase in scores, then there will be an increased misdetection rate.

Our data indicate a decrease in the diagnostic quality with regard to specificity and sensitivity of the SOFA score, which occurs shortly after onset of treatment. Hence, aspiration pneumonia, which is often observed at the onset of SAH treatment, can be detected rather reliably by the SOFA score a septic condition [18]. However, to enable the early diagnosis of sepsis, the investigator’s clinical views and synopsis of various scores and laboratory parameters should be highlighted. Our results should also be critically reviewed in light of increasingly semiautomated data processing. Inexperienced clinicians may make the misleading assumption of sepsis that requires therapy, thus leading to overtreatment with antibiotics. Therefore, we suggest that the purposeful collection of each SOFA or SIRS module should be critically reviewed [12]. Due to prolonged therapy, critically ill SAH patients can have a high risk of nosocomial infection, as in our collective, ventilator-associated pneumonia (VAP), as well as urinary tract infections (UTIs) and catheter-associated infections (e.g., central line-related bloodstream infection [CLABSIs]), were predominant [18,19]. Thus, as observed in our cohort, the majority of SAH patients demonstrated more than one septic infection. In particular, the detection of such a secondary infection should be achieved as soon as possible due to possible infectious source control and the need for antibiotic therapy. However, as our data suggest, the SOFA score exhibits insufficient sensitivity and specificity among critically ill SAH patients, especially in the presence of CVS. The increase in the SOFA score that is required for the detection of sepsis is not very possible because the circulatory and neurological modules had frequently already reached maximum score values. In addition, due to mechanical ventilation, one or two points in the respiratory module were already being continuously detected. In this respect, the question of diagnostic quality arises in prolonged intensive care stays with the uncertainty of when a new increase in points should be considered as a new detection of sepsis and no longer as a worsening of the condition of the same episode of sepsis.

The occurrence of CVS exhibits a clear correlation with the severity of SAH, however, a therapeutic treatment to prevent ischemic stroke in patients with cerebral vasospasm is still missing [20,21,22]. Therefore, it may also be assumed that in the future, a high mean blood pressure will be applied for the supposed optimization of cerebral perfusion and that catecholamines, which are regularly necessary to achieve such a target, will impair the performance of the SOFA score.

However, the connection between cerebral vasospasm and inflammation, which has been repeatedly discussed, remains an exciting aspect; in addition, the extent to which a systemic infection influences the presence of cerebral vasospasm has not yet been adequately investigated. A novel approach to the detection of sepsis in critically ill SAH patients is not yet available, which is surprising in view of the significantly increased mortality risk of SAH patients suffering from sepsis [23,24].

In contrast to the recently implemented sepsis-defining SOFA score, the previously used SIRS score claims to reflect a septic state with suspected bacteremia by a scoring value ≥ two criteria. In the study that we performed; we detected a slight advantage in terms of detecting sepsis. This finding accentuates the proneness to error of the SOFA scoring in impaired modules. In contrast to the SOFA score, the differently defined SIRS criteria are otherwise impaired. Under mechanical ventilation, the assessability of respiratory rate is eliminated, but the body temperature, heart rate and leukocytosis criteria are still retained. However, for the diagnostic quality of the SIRS score, it should be considered that body temperature can also exhibit febrile temperatures based on a neurogenic fever without a septic event occurring [25,26].

The sepsis 3 criteria claim to be globally applicable and likewise should be assessable with basic laboratory parameters under simple conditions. However, numerous other diagnostic options are available within the framework of modern critical care medicine. In particular, laboratory parameters of infection such as C-reactive protein (CRP), procalcitonin (PCT) and interleukin 6 (IL-6) are able to provide valuable information on the presence and progression of infection. In addition, clinical parameters such as the increased presence of tracheal mucus and its tint and consistency (for example, purulent, viscous) can be valuable additions to the Horovitz index of the SOFA score. The detection of a urinary tract infection can be accelerated by regular urine testing (for example, as ready-to-use test strips).

Our data suggest the limited usability of the SOFA score in critically ill SAH patients. Among 141 SOFA-positive events, the majority (89) proved to be false detections. We were able to demonstrate a significant correlation with therapeutic blood pressure elevation and the need for deep sedation with concomitant low GCS. Therefore, the application of the SOFA score in critically ill SAH patients should always be considered in light of its possible limitations.

## 5. Conclusions

The SOFA score creates a new era of sepsis definition with predominantly positive study results. However, in special patient populations, such as critically ill SAH patients, it possesses certain limitations. In particular, it is very important to differentiate between CVS and sepsis. Therefore, non-neurology tools often lead to misinterpretation in sepsis detection.

## Figures and Tables

**Figure 1 jcm-11-04229-f001:**
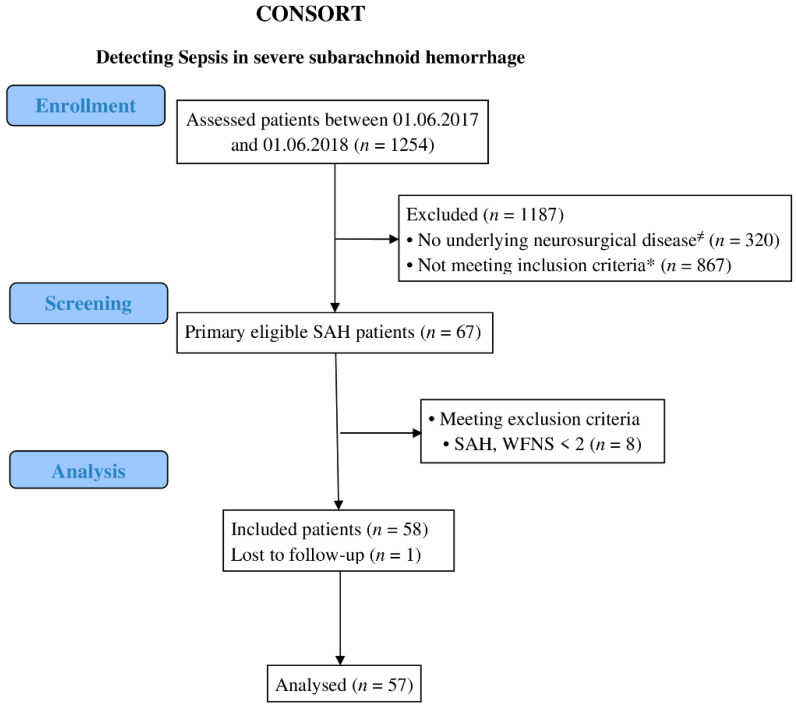
Consolidated Standards of Reporting Trials (CONSORT) diagram of patients included in the study. Diagram of the inclusion process and the reasons for exclusion. Abbreviations: SAH, subarachnoid hemorrhage; WFNS, World Federation of Neurosurgery clinical severity of subarachnoid hemorrhage. ≠ Intensive care admission due to primary neurological disorders * Intensive care unit admission due to neurosurgical diagnosis in the without presence of subarachnoid hemorrhage.

**Figure 2 jcm-11-04229-f002:**
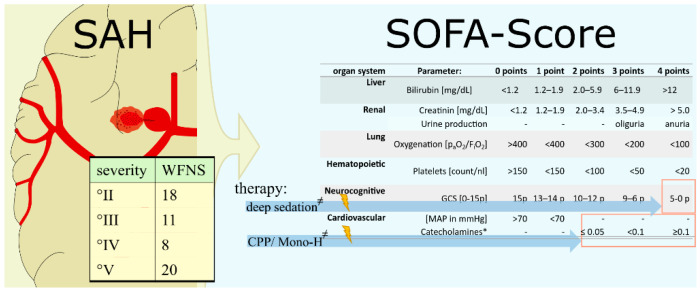
Severity of subarachnoid hemorrhage and influence of sepsis-related organ failure assessment score. Grade of subarachnoid hemorrhage according to the World Federation of Neurosurgery. The possible influences of the corresponding therapy on SOFA score. Abbreviations: CPP, cerebral perfusion pressure; CVS, cerebral vasospasm; dL, deciliters; FiO2, inspiratory oxygen fraction; GCS, Glasgow Coma Scale; MAP, middle arterial pressure; mg, milligram; mmHg, millimeters mercury; Mono-H, therapy strategy of cerebral perfusion protection by raising the mean arterial pressure >90 mmHg; nl, nanoliter, p_a_O_2_, arterial oxygenation; p, points; SAH, subarachnoid hemorrhage; SOFA, sepsis-related organ failure assessment; WFNS, World Federation of Neurosurgery. * Initial SOFA Score publication lists dobutamine as the first-choice catecholamine. Moreover, this approach has been abandoned in favor of using norepinephrine as the first-choice catecholamine. ^≠^ Significant impairment of the SOFA score with respect to false positive sepsis detections.

**Figure 3 jcm-11-04229-f003:**
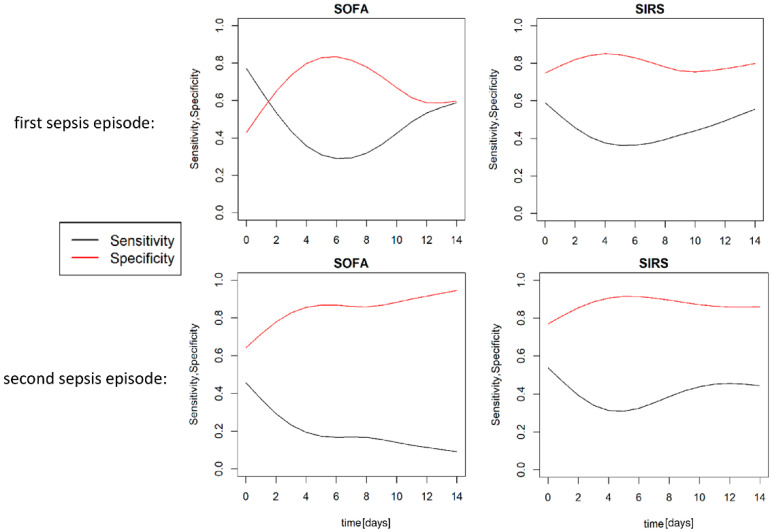
Sensitivity and specificity of sepsis detection in severe subarachnoid hemorrhage. Detection of sepsis occurrence via the sepsis-related organ failure assessment and systemic inflammatory response syndrome criteria for the first sepsis episode during critical care treatment for severe subarachnoid hemorrhage, as well as for a subsequent sepsis episode. Reports of the sensitivity and specificity of the recommended sepsis definitions according to sepsis-2 and sepsis-3 criteria.

**Table 1 jcm-11-04229-t001:** Clinical characteristics according to the septic episodes.

	Total	No Sepsis Detected	1 Septic Episode	≥2 Septic Episodes
Patients, *n*	57	10	21	25
Sex (male), *n*	27 (47%)	4(40%)	11(52%)	12(48%)
age, y	56 (29-92)	52 (24)	59 (20)	57 (14)
ICU stay, d	17 (17.3)	10 (4.9)	21 (9.8)	35 (10.9)
GCS admission, p	8.0 (5.7)	12.2 (5.2)	6.9 (5.7)	7.0 (5.3)
**SAH treatment ^≠^**				
operative Clipping	22	3	9	9
endovascular Coiling	33	5	12	16
**infection site**				
pneumonia	60	0	20	40
CLABSI	13	0	1	12
Urinary tract infection	7	0	0	7
ventriculitis	2	0	0	2
**Outcome parameters**				
Antibiotic treatment *	126	2	37	87
CVS	40 (70%)	3 (30.0%)	13 (62%)	24 (96%)
death	9 (15.8%)	1 (0.1%)	7 (33.3%)	1 (0,1%)

Patient clinical characteristics differentiated by the frequency of observed septic episodes. Data are presented as the mean with standard deviation. Abbreviations: CVS, cerebrovascular spasm; CLABSI, central line-related bloodstream infection; d, day; GCS, Glasgow coma scale; ICU, intensive care unit; p, points; SAH, subarachnoid hemorrhage; SD, standard deviation; y, year. * Number of anti-infective treatments that were administered. ^≠^ Two patients were not treated due to unfavorable prognoses.

**Table 2 jcm-11-04229-t002:** Sepsis detection via SOFA score in SAH.

	SOFA				SIRS			
	Day 0	Day 1	Day 6	Day 10	Day 0	Day 1	Day 6	Day 10
sensitivity	77.1%	64.9%	28.9%	42.4%	59.0%	52.1%	36.3%	44.0%
specificity	42.9%	54.4%	83.3%	66.8%	74.8%	78.6%	82.9%	75.5%
prevalence	0.0%	4.4%	18.1%	20.4%	0.0%	4.3%	21.0%	22.2%
PPV	0.0%	6.2%	27.8%	24.6%	0.0%	9.9%	36.1%	33.9%
NPV	100%	97.1%	84.1%	81.9%	100%	97.3%	83.0%	82.6%

Performance of sensitivity, specificity, negative predictive value, and positive predictive value regarding the ability of the SOFA score to correctly detect the occurrence of the first septic episode in critically ill patients suffering from subarachnoid hemorrhage (SAH). Abbreviations: SOFA, sepsis related organ failure assessment; SIRS, systemic inflammatory response syndrome; NPV, negative predictive value; PPV, positive predictive value.

**Table 3 jcm-11-04229-t003:** Regression analysis of correlation of false positive SOFA detections to impaired Score modules.

SOFA Score Module	Clinical Correlate	*p*-Value
Glasgow coma scale	intubation/sedation	<0.01
Hypotension/circulatory	catecholamine consumption	<0.01
Hypotension/circulatory/Glasgow coma scale	attributable to cerebrovascular spasm	<0.01

Results of regression analysis regarding the underlying cause of false positive SOFA score driven sepsis detections.

## Data Availability

Data cannot be shared publicly. The datasets generated and/or analyzed during the current study are not publicly available due to national data protection laws but are available upon reasonable request from the corresponding author, or via the data protection officer of the University Hospital Frankfurt (datenschutz@kgu.de).

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
