# Peer review of "Detecting Sepsis in Patients with Severe Subarachnoid Hemorrhage during Critical Care"

_jcm, 2022, doi:10.3390/jcm11144229_

Round 1
Reviewer 1 Report
Thank you for giving me the opportunity to review the article above. The topic is interesting and merits publication. The difficulties in diagnosing sepsis based on current sepsis definition in this patient population are clearly shown.
The article is well written, but I do have one minor comment:
Could the authors further comment in the discussion section on how to diagnose sepsis in this patient population if SOFA score in not useful? Rapid recognition and initiation of antibiotic and other sepsis related treatment is remains paramount in this patient population (as shown by high prevalence of sepsis), but which tools can we use to identify patients with sepsis?
Reviewer 2 Report
There is a general problem in the way Tables and Figures are reported in the text. The table provided as supporting material is the same as Table 1. Figure 2 is not very understandable and it looks as it does not belong to where it is cited as it does not relly present any result. I would suggest to totally revise figure 2 or to singnificanlty improve the legend.
Correlations results are very interesting but only presented in the text, I suggest to present them in a table/figure or at least to report them as a supplementary file.
I would add the WFNS score on admission to the Table 1.
Methods: I suggest to describe more into details how were the sepsis episodes determined, as this is important for the comparison with the sofa score.
Please revise this sentence in the discussion: "This finding accentuates the proneness to error of the SOFA scoring in impaired modules, in contrast to the SOFA score, the differently defined SIRS criteria are otherwise impaired."
Reviewer 3 Report
Thank you for the opportunity to review this paper.
I think this is a very meaningful article, becuase it is the first study to evaluate the applicability of the SOFA score in the detection of sepsis in SAH patients. Nevertheless, there are some areas that need to be revised considerably.
First of all, It is necessary to present the statistical results in Figure 3 as a table. In addition, NPV, PPV and P value as well as sensitivity, specificity should be presented. This results are clinically important.
Second, it is necessary to present the NPV, PPV, sensivity and specificity of the SOFA score without the GCS and/or hypotension score. Moreover, comparison of NPV, PPV, sensivity and specificity between the SOFA score without the GCS and/or hypotension score, and the original SOFA score could provide more meaningful information.
Minor
The meaning of “assessed patients” in Figure 1 is unclear. In the Figure 1, the study period and criteria for which the first 1254 people were screened should be presented.
In Figure 3, the units of time on the X-axis should be presented.
